# Long-Term Results of Joint Arthroplasty with Total Prosthesis for Trapeziometacarpal Osteoarthritis in Patients over 65 Years of Age

**DOI:** 10.3390/geriatrics6030065

**Published:** 2021-06-29

**Authors:** Miguel Angel Martin-Ferrero, Jose Maria Trigueros-Larrea, Elsa Martin-de la Cal, Begoña Coco-Martin, Clarisa Simon-Perez

**Affiliations:** 1Orthopedic and Trauma Service, Hospital Clínico Universitario, University of Valladolid, 47003 Valladolid, Spain; ferrero@cir.uva.es (M.A.M.-F.); josemaritrigueros@yahoo.es (J.M.T.-L.); 2Funge Research Unit, University of Valladolid, 47002 Valladolid, Spain; mmartinferr@saludcastillayleon.es (E.M.-d.l.C.); mbegocococom@gmail.com (B.C.-M.)

**Keywords:** arthroplasty, trapeziometacarpal, osteoarthritis, older population

## Abstract

Trapeziometacarpal osteoarthritis (TMCOA) is a highly prevalent disease in the older population. Many different types of surgical treatments are possible, depending on the degree of joint involvement, the personal and professional circumstances of the patient and the preferences of the orthopedic surgeon. This paper evaluated the clinical and radiological results of consecutive cohorts of patients over 65 years old treated with total joint arthroplasties (TJA) of the ball and socket type (B&S) for TMCOA, with a minimum of 10 years follow-up. The survival rate (Kaplan–Meier) of the functional prostheses at 10 years was 92.2% (95% CI (89.1%, 96.1%). These functional arthroplasty patients, after 10 years of follow-up, showed little or no pain, good function and good key pinch, without radiological alterations. TJAs of the B&S type are a long lasting, effective and reliable alternative to surgical treatment of TMCOA in patients over 65 years of age, when they are performed with the patient selection criteria and surgical technique described throughout this study.

## 1. Introduction

Trapeziometacarpal osteoarthritis (TMCOA) is a highly prevalent disease, with up to 25% of the adult population suffering from this degenerative process. It is much more frequent in women than in men, with reported prevalences ranging from 10 to 15:1, depending on the study. Although not all patients suffer pain, 55% of patients with combined trapeziometacarpal and scapho-trapezio-trapezoid osteoarthritis complained of basal thumb pain. Pain is also not related to the radiological stage of the disease [1]. When estimating the associated sex, age and intensity of osteoarthritis, observations by Sodha et al. in 2005 [2] found that up to 50% of women over 70 years of age had destruction of the carpometacarpal joint. The influence of a specific manual occupation has not been demonstrated as a reliable cause, but repetition of the gripper movement, which is very common when painting, washing clothes, opening a door with a key or opening screw-locked plugs, performed by housewives and some populations of manual workers, seems to predispose to this pathology [3]. The concomitant effect of joint laxity, which is almost always present in women between the ages of 15 and 35, and which seems to have an insufficiently contrasted hormonal background, may be an added cause, as has also been found in other types of diseases causing ligamentous laxity [4].

When treating this pathology, it is very important for the patient to have, from the beginning, the necessary information to understand the natural history of the condition and the different therapeutic options, with their benefits and risks. It is important to understand that the degenerative process is not reversible, but that it is possible to control the symptoms with pain relievers, splints and physiotherapy and intelligently adapting activities at home and at work. Only when medical treatment has failed, should surgical treatment be performed. Many different types of surgical treatments are possible, depending on the degree of joint involvement, personal and professional circumstances of the patient and the preferences of the orthopedic surgeon.

The most common surgical treatments when TMCOA is already established are: Trapeziectomy [5], its relative simplicity and the few complications inherent in the technique have meant that, at present, it remains the benchmark against which most procedures are compared. The proximal migration of the thumb metacarpal, which decreases functionality, strength and aesthetic appearance, relegates this technique to very elderly patients with little functional demand. Trapeziectomy and tendon interposition, to try to avoid the collapse between the first metacarpal and the scaphoid was described by Froimson, in 1970 [6]. Subsequently, numerous types of interposition tissues between the metacarpal and the scaphoid have been described. The studies performed with these techniques have little evidence and only allow us to conclude that this procedure offers a result similar to that of trapeziectomy. Burton and Pellegrini [7] were the first, in 1986, to describe the technique of trapeziectomy and ligamentous reconstruction with tendon interposition (LRTI) with half of the flexor carpi radialis tendon. The number of papers describing series of patients with this procedure and different technical variations is unmanageable. Although it seems that this technique decreases the collapse between the first metacarpal and the scaphoid, improving the length of the thumb, there is controversy in the literature, since some meta-analyses show that the results are not superior to those of trapeziectomy alone and there are a higher number of complications due to the greater technical complexity [8]. Trapeziometacarpal arthrodesis was described in the mid twentieth century [9]. A small number of authors discuss its indication in young patients with heavy manual work because it maintains the length of the thumb and the strength of the grip. However, there is a reduction in the mobility of the thumb and the results are not very predictable due to the large number of complications, especially nonunion, which can occur in 39% of cases [10].

In order to solve the collapse of the first metacarpal of the thumb that occurs after simple trapeziectomy, the mobility problems and the difficulty in achieving fusion of the arthrodesis, arthroplasty with interposition implants began to be used in the 1960s of the last century. The first described interposition implants were made of silicone [11], but their instability and synovitis caused by silicone particles detached by friction led to their abandonment. Other interposition implants have not performed better.

More recently, total joint arthroplasty (TJA) has been used to treat TMCOA. There are many types of TJA, but the ball and socket type (B&S) has a different behavior from other prostheses and is the only type for which multiple long-term studies have been published. The first-generation B&S TJAs, designed by De la Caffinere [12], show acceptable long-term survival, but with unacceptable loosening rates [13]. The second generation of B&S TJA [14] is modular, non-cemented, and unconstrained and is the most commonly used procedure at present. Several papers have reported good long-term survival results (over 90%), with a much lower rate of loosening [15].

The aim of this study is to focus the investigation on an older population, to evaluate the survival results of a cohort of consecutive patients over 65 years old treated with B&S TJA for TMCOA in the thumb, with a minimum of 10 years follow-up.

## 2. Materials and Methods

A follow-up study was performed at our hand unit with 64 TJAs in 60 patients over 65 years of age who had undergone surgery between May 1999 and May 2010. The consecutive patients included in the study have undergone surgery using a modular, non-cemented, hydroxyapatite-coated, unconstrained Arpe^®^ implant (Biomet France, Plateau de Lautagne, Valence, France).

The data on age, sex, and degree of osteoarthritis and implant size were recorded. Written informed consent was obtained from all the patients, and approval from the local Hospital Research Ethics Committee was acquired.

### 2.1. Inclusion and Exclusion Criteria

We started using the original inclusion criteria of De la Caffiniere [12]; later, based on our experience, we adjusted them slightly and these adjusted criteria were used in the study. Patients must have osteoarthritis of Eaton’s grade II and III and some grade IV [16], be aged over 65 years, and may engage in light or medium but not excessively heavy manual tasks. Patients were excluded if they had marked ulnar instability or fixed hyperextension of the metacarpophalangeal joint (MCPJ), severe osteoarthritis of the scapho-trapezio-trapezoid joint, severely dysplastic trapezius, rheumatic diseases, or heavy manual work requirements such as hammering, drilling, or strong mechanics.

### 2.2. Changes in the Surgical Procedure

From May 1999 to April 2004 a single surgeon (MMF) performed the prostheses using a lateral–palmar approach (original technique) recommended by the designer of the Arpe prosthesis [14]. In May 2004, MMF introduced important changes in the original technique that facilitated the surgical procedure, and this procedure is the one we continue to perform (current technique). The technique adopted is a purely dorsal surgical approach, performing a more extensive resection osteotomy of the base of the first metacarpal to improve access to place the cup. Osteotomy of the trapezium is no longer performed, only resecting peripheral osteophytes, which preserves the distal subchondral bone for improving the primary support of the cup. The current surgical technique is described in detail in a previous publication [17]. After surgery, a plaster cast is applied with the thumb in a functional position and left in place for 3 weeks. In most instances, patients rehabilitated themselves by following a comprehensive exercise program, but a specialist in hand therapy was available if required.

### 2.3. Follow-Up and Clinical Assessment

Clinical and radiological assessments were performed preoperatively and then postoperatively at 3 months, 5 years, and 10 years. The clinical examination consisted of measurement of the range of motion of the thumb using the criteria of International Federation of Societies for Surgery of the Hand [18], and the thumb opposition using the Kapandji method [19]. Key pinch strength was measured using a pinch gauge (B&L Engineering, Alimed Inc., Dedham, MA, USA).

The radiological examination comprised posterior–anterior and oblique radiographs. Preoperative radiographs were classified by the Eaton criteria [16], and postoperative radiographs were evaluated to observe the following factors: implant component alignment, subluxation or dislocation, implant loosening and/or subsidence, and ectopic calcifications.

The patient fulfilled two objective tests: The pain level was measured using a visual analogue scale (VAS) (ranging 0–10), and patient satisfaction and function was measured using the Disabilities of the Arm, Shoulder, and Hand (DASH) questionnaire translated to Spanish [20]. All the patients completed the DASH questionnaire at 10 years post-surgery.

### 2.4. Definition of Outcomes

We developed criteria to scale the implant performance that are shown in Table 1.

### 2.5. Statistical Analysis

Parametric data are presented as the means with standard deviation (SD). Non-parametric data are presented as medians with interquartile range (IQR) for continuous variables or as percentages for dichotomous variables. Kaplan–Meier analysis was performed to assess the distribution of outcomes. Differences between the preoperative and follow-up measurements of clinical numerical variables were assessed using post hoc Student *t*-tests after analysis of variance. Fisher’s exact test was applied to check the association between outcome dichotomous variables and the two subgroups given by the implant date. *p* values lower than 0.05 were considered statistically significant.

## 3. Results

From May 1999 to May 2010, a total of 289 TJAs for TMCOA in 252 patients (37 bilateral) were operated on with Arpe prostheses in our hand surgery unit. The target of this study was patients over 65 years of age at the time of surgery, which involved a total of 64 prostheses placed in 60 patients. During the follow-up period seven patients (with seven prostheses) had died before of the 10-year revision. It was not possible to locate two patients (two prostheses). Four patients (with four prostheses) were unable to attend the outpatient clinic review and were interviewed by telephone (they are included in the clinical results). Therefore, 55 of the 64 joints (86%) completed the 10-year follow-up (median: 10.2 y; 25th, 75th percentile (10.0 y, 11.5 y)

Fifty-seven patients were women and three were men. The incidence was approximately the same in the dominant (53%) and non-dominant hands (47%). Ages ranged from 66 to 83 years, with an average age of 72.5 years, SD 7.2 (56 TJAs were from 66 to 76 y). Physical activities were light in five hands (8%); moderate (housewife, hairdresser) in 49 (79%), medium-hard in four (6.5%) and no information was available for the remaining four. The etiology in all patients was osteoarthritis. Eleven thumbs (17%) were of Eaton stage II, 46 (72%) were stage III, and seven (11%) were stage IV. Twenty-nine surgeries (45%) had coexisting pathologies with the TMCOA, with carpal tunnel syndrome and trigger finger being the most frequent, and these were operated on at the same time as the prosthesis. The most frequent cup and stem sizes were nine, and more than 95% of the necks were angulated and medium size.

### 3.1. Short-Term Results and Complications

Sixty-one thumbs (95%) had recovered at week six of postoperative review. Early postoperative complications included: Two linear bone fractures of the trapezium that required immobilization for four weeks instead of three. One of the first TJAs had a false pathway of the stem, which remained in situ and functional without evidence of loosening in the ten-year follow-up review. Two early dislocation (before 6 months): one solved with close reduction, and one that was surgically repaired, both were functional at the latest follow-up. Six patients had dorsal paresthesia or dysesthesia in the thumb that disappeared in the first year after surgery. One patient experienced transient complex regional pain syndrome type I, which resolved in the first six months. Two cases had a superficial suture reaction with temporal dehiscence. There were no deep infections.

### 3.2. Survival

The survival rate of these patients over 65 years at 10 years was 92.2% (95% CI (89.1%, 96.1%)) and the Kaplan–Meier curve is shown in Figure 1. The 26 prostheses performed using the current technique showed a greater survival than the average of the series, being 94.1% (95% CI (89.7%, 97.2%)), although this was not statistically significant.

Functional TJAs after 10 years of follow-up showed little or no pain VAS preoperative: 8 (SD 1.3); VAS at 10 years: 1.3 (SD 1) (*p* < 0.001) and good function DASH preoperative: 61 SD 9.5; DASH at 10 years: 16 SD 12.6 (*p* < 0.001). Similar figures were observed for the Kapandji’s score (91% thumbs scored over 9) and radial abduction degrees (89% were over 30°). The key pinch was 2.9 SD 1.5 kg preoperatively and 3.9 SD 1.8 kg at 10 years.

Radiological assessment was performed in 45 (92%) of functional joints. Of these, 39 (87%) showed good implant integration without any loosening. Six (13%) presented some ectopic calcification and slight radiolucency, but these were not associated with any adverse symptoms. The mean distance from the cup bottom to the scaphoid trapezium trapezoid joint was 4.4 mm (SD 1.6). Radiological implant positioning was better in the 26 prostheses that had been inserted using the current technique versus the original technique, and the differences were statistically significant. There was a lower incidence of oblique misalignment of the stem in PA XR view i.e., 12 (41%) for the original technique vs. two (8%) for the current technique (*p* < 0.001); implant subluxation occurred in 7 (22%) vs. 3 (6%), respectively, (*p* < 0.05); and cup subsidence occurred in 6 (21%) vs. 1 (4%), respectively, (*p* < 0.005). Figure 2.

### 3.3. Late Complications

Late complications included cup loosening in three cases (5.4%). One was successfully treated with a cup revision with autologous bone grafting and was functional at the latest follow-up, the other two will be described in the failed prostheses section. Two prostheses (3.6%) were late dislocations that remained dislocated at the final follow-up (detailed among the failed prostheses). A prosthesis that was painful and non-functional is also described next in the failed prostheses section. No complete loosening of the stem was present in any case. Two implants showed partial radiolucency proximally in the stem without clinical repercussion.

### 3.4. Description of Failed Prostheses at Ten Years Follow-Up

A 71-year-old woman with small a trapezium, was operated on 14 June 2001, without any adverse event during surgery, and with no alterations noted in XR postop. At 5 years a painful cup loosening appeared. We then performed removal of the cup and neck, trapeziectomy, and LRTI. The prosthetic stem was left inside the metacarpal because it was well integrated. In the last follow-up (16 October 2013) the patient had a correct mobility, without pain. The second failed prosthesis occurred in a 68-year-old woman, with a normal trapezium and joint Laxity. Surgery on 10 January 2002 was not perfect, with excessive resection osteotomy of the trapezium. Postop XR shows the excessive osteotomy and some obliquity of the stem, which led to a subluxation. Pain and reduction of mobility began 8 years after surgery. The patient rejected reparative surgery. In the last follow-up review on 8 February 2012, we found a VAS score of 6, and a DASH of 50. The third failed prosthesis occurred in a 75-year-old woman, with a small trapezium and MCPJJ joint laxity. The surgery was performed on 17 October 2003. In the postop XR we observed the cup in a bad position, with a dorsal tilt. At eleven months the patient appeared in casualty with a prosthetic dislocation. As it was not painful, she rejected restorative surgery. In the last follow-up on 29 January 2014 the prosthesis remained dislocated but was not painful. The fourth occurrence was in a 67-year-old woman, with a small trapezium and articular laxity. The patient was operated on 22 December 2006, without incident. The postoperative X-ray showed the correctly positioned implant. After six months the patient underwent knee OA surgery and she had to use crutches for several months. At a year and a half of evolution, she presented with a dislocation of the thumb prosthesis. She refused reconstructive surgery. At the last follow-up on 21 December 2016 the prosthesis was still dislocated, but with little pain. The fifth occurrence was in a 78-year-old woman with a small trapezium. The surgery date was 28 January 2010 and the operation was without incident. The postoperative X-ray showed no alterations. The prosthesis retained its functionality until nine years of evolution when, after a traumatic thumb twisting, she began to feel pain and the cup loosened. We performed cup and neck extraction, with trapeziectomy and LRTI. The prosthetic stem was left inside the metacarpal because it was well integrated. At the last follow-up on 10 October 2020 the patient showed functional thumb mobility with no pain.

It is remarkable the frequency of small trapeziums (defined as height in the radiographic projection PA < 9 mm) in the failed cases (4 out of 5) was much higher than in patients with functional outcomes (9 out of 50).

## 4. Discussion

This is a study of a considerable size of 64 TMC joint Arpe^®^ prostheses in a consecutive series of patients older than 65 years with a follow-up of at least 10 years, with a low attrition. In addition, there are two other considerations to be noted. First, precise surgical indication criteria were established before the patients were included; second, we have defined clearly the characteristics that functional and failed prostheses must have in evaluating the results. After these considerations, our experience with the Arpe^®^ prosthesis for the treatment of TMCOA in patients over 65 years of age is quite satisfactory, considering that 92% (Kaplan–Meier survival rate) of the prostheses remain functional and without pain.

The present study can justifiably be criticized for studying the practice of two surgeons undertaking this type of operation over a long period of time, rather than the outcomes achieved by more surgeons. Another weakness is that the study design did not reduce the potential for bias, because the treating surgeons and members of our staff carried out all the preoperative and follow-up assessments. Although there is a risk that subconscious bias affected the outcome measures, the subjective outcomes were free from observer bias as the patients completed the VAS and DASH objective questionnaires on their own.

Initially, we used the inclusion and exclusion criteria established by De la Caffiniere and Aucouturier (1979) [12], and then we adjusted them as we gained experience. We introduced for the first time in the literature classification criteria (Martin Ferrero, 2020) [17] to consider arthroplasty as functional or failed (Table 1). We’d like that those should be tested and discussed in subsequent studies of prostheses results.

TMC prostheses in general, without going into detail, have been questioned by some authors because of their price and because the long-term complications are often higher than trapeziectomy [21]. However, it seems clear that trapeziectomy, which is a “mesenchymal arthroplasty” after having performed a trapezius excision, cannot be a way of functioning of the peculiar biomechanics of this joint, explaining the poor functional quality of the thumb clamp [22]. In addition, there are more and more studies comparing the functional results of B&S TJAs and trapeziectomies in their different forms, and prostheses clearly outperform trapeziectomies in postoperative recovery time, functionality and aesthetic appearance of the thumb [23].

The long-term survival of prostheses has also been questioned, but there is a growing number of studies of the long-term evolution of second-generation B&S TJAs [15,17,24,25,26,27,28,29,30] that report excellent survival (89–96%) and much lower loosening rates (4–7%) in these prostheses than that of first-generation prostheses which have survival rates ranging from 76% to 82% and loosening rates of up to 40% [13,31,32]. However, second generation current prostheses may dislocate at a frequency of approximately 5% [24], because they are not constrained. It is also worth noting the similarity of the survival rates of the current second-generation implants to those reported by Allami et al. (2006) [33] on the 10-year survival of TJAs of the hip (93%), which is the standard reference in orthopedic prostheses.

Vermeulen et al., in an evidence-based study, concluded that at this time, no surgical procedure has proven superior to others. However, they postulated, based on the good results of the B&S type of TJA, that there could be differences between the various surgical procedures [8].

Based on our experience, we changed the technique of how to place and fix the Arpe^®^ prosthesis to bone and thereby tried to improve the results and survival. The current dorsal approach permits better access to the distal trapezium to improve the cup positioning, allows a better introduction and control of the direction of the stem into the first metacarpal, and diminishes the incidence of radial neuritis. Regarding the positioning of implant components, the results of the current technique really improved on those of the designer’s original technique, with statistical significance (*p* < 0.005), and we would expect that the survival rate will improve with the current technique even beyond the ten years that we have investigated in this study [17].

If we compare this study of older patients over 65 y with the previously published series mentioned above in this discussion, all the data are quite similar referring to functionality and survival [15,17]. This means that prostheses can be placed in patients over 65 years of age, with the same confidence as in younger patients.

When analyzing the causes of prostheses failure, we find two main causes. A small trapezium was present in 80% of the failed prostheses and only in 18% of functional prostheses. Brutus y Kinen, in 2004 reported similar findings [34]. This does not prevent the use of prosthesis in those cases, but a special caution must be adopted during previous XR studies and in the surgical procedure in patients with a small trapezium. Figure 2. The preoperative first metacarpal adduction and the laxity of the metacarpo–phalangeal joint associated with TMCOA also contribute considerably to the failure with prosthetic dislocation or subluxation, as also was reported by Badıa and Sambandam (2006) [35]. Therefore, the prosthesis and these MP associated alterations must be treated at the same surgery to avoid complications.

In case of failure (loosening, dislocation) without reparative options, a trapeziectomy and LRTI was performed after removing the prosthesis. Sometimes the stem is left in place if it is fully integrated. The results after this procedure are similar to those cases of primary trapeziectomies and LRTI, as has also been pointed out by Cooney [36].

## 5. Conclusions

This series of patients with long follow-up has demonstrated that Arpe^®^ prosthesis is a long lasting, effective and reliable alternative for surgical treatment of TMCOA in patients over 65 years of age, if it is performed with the criteria of surgical indication and surgical technique described throughout the study.

## Figures and Tables

**Figure 1 geriatrics-06-00065-f001:**
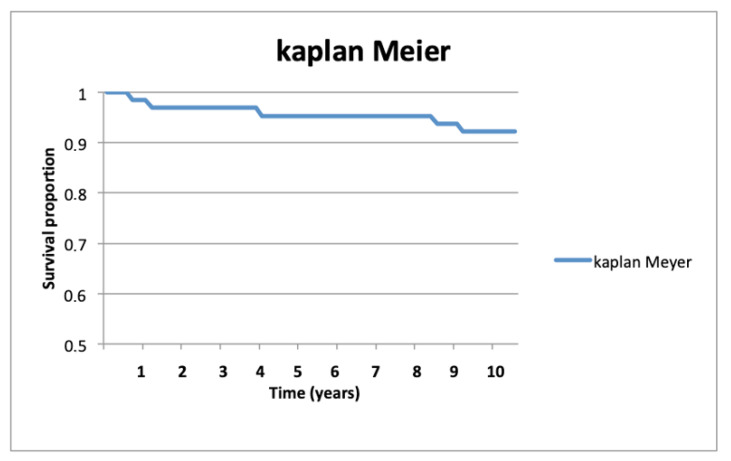
Kaplan–Meier curve at 10 years.

**Figure 2 geriatrics-06-00065-f002:**
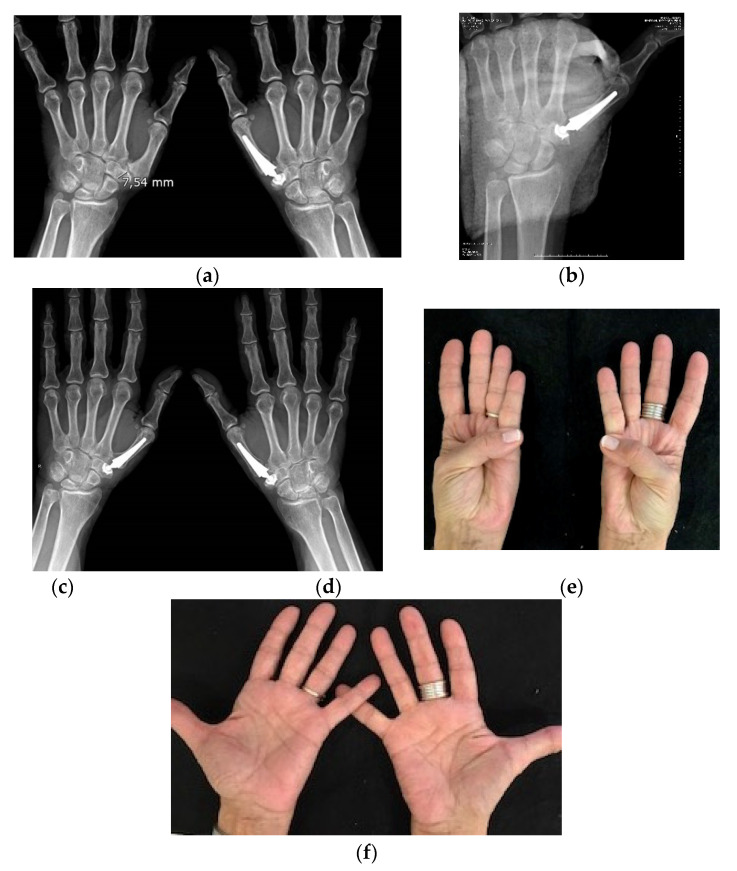
(**a**) 67 year-old patient with bilateral trapeziometacarpal arthrosis. She has had an Arpe prosthesis on the right side for four years. She underwent surgery for prosthesis placement on the left side, where a small trapezium of 7.5 mm was observed; (**b**) Postoperative X-ray control. In a small trapezium there is enough space to place a cup with certainty if good planning and surgical technique are used; (**c**,**d**) AP and oblique X-ray control of both hands with 11 years of evolution on the left side and 15 years of evolution on the right side; (**e**,**f**) Thumb mobility at the last follow-up.

**Table 1 geriatrics-06-00065-t001:** The prosthesis was considered to be functional if the patient has no pain and no alterations in mobility, functionality and radiology and could use the hand normally for activities of daily living. The prosthesis was considered as failed if at least one of the major alterations occurred. The prosthesis was also considered failed if at least three or more minor alterations were present.

	Functional TJA	Failed TJA	Failed TJA
		(1 Major Alteration)	(3 Minor Alterations)
Mobility	radial abduct. > 30°	radial abduct. ≤ 20°	radial abduct. 20–30°
	kapandji score > 8	kapandji score ≤ 6	kapandji score 7–8
Objective test	VAS < 3	VAS ≥ 5	VAS 3–4
	DASH < 30	DASH ≥ 40	DASH 30–40
Radiology	without adverse events	complete loosening	radiolucency of components
		dislocation	ectopic calcifications
			subsidence
			subluxation

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
