# Peer review of "Long-Term Results of Joint Arthroplasty with Total Prosthesis for Trapeziometacarpal Osteoarthritis in Patients over 65 Years of Age"

_geriatrics, 2021, doi:10.3390/geriatrics6030065_

Round 1
Reviewer 1 Report
It would be more impactful to include minimal clinical important differences (MCID) when reporting outcome scores.
Does fixation (press-fit versus cement) affect long-term or short-term outcomes?
Author Response
It would be more impactful to include minimal clinical important differences (MCID) when reporting outcome scores.
Answer.- All the clinical, radiographic and objective test values are perfectly defined in table 1 (line 149), what we have done is to apply these values to each patient. Taking into account the above, the prosthesis will be functional or failed. From line 200 onwards the short term and long term results (survival) are specified.
Does fixation (press-fit versus cement) affect long-term or short-term outcomes?
Answer.- There are previous studies (De la Caffinniere) with cemented arthroplasties in which the fundamental problem was loosening. For this reason, all our patients have had cementless prostheses (pres-fit).
Than you very much for your time and interest
Reviewer 2 Report
This study discusses the long-term experience of two surgeons replacing the trapeziometacarpal joint with a total joint prosthesis. The study is of value because there are only few reports with follow-up times of >10 years and survival rates of >90% in the literature.
Overall the study is well presented. A motivation why the focus of this study has been placed on the elderly (i.e. subjects >65 yrs.) is missing and should be added.
There are a few smaller issues that should be corrected before publication:
- There are three abbreviations for trapeziometacarpal osteoarthritis: TMOA, TMCOA, TMCA. Please choose one and be consistent
- The first three sentences of the introduction require literature
- It is never clearly stated that the subjects were >65 years of age at the time of surgery. Please do so.
- Did I miss the definition of “CTS” in line 179?
- The authors call out the frequency of small trapeziums in the failed group, but not in the successful group (lines 266-268)
Spelling:
- Line 20: patient cohort
- Line 22: Kaplan-Meier
- Line 89-90: show acceptable long-term survival
- Line 161: considered
- Lines 200-202: Please correct grammar (e.g. put VAS ranges in parentheses)
Author Response
There are three abbreviations for trapeziometacarpal osteoarthritis: TMOA, TMCOA, TMCA. Please choose one and be consistent
Answer.- All of these corrections have been made in the modified manuscript to TMCOA
The first three sentences of the introduction require literature
Answer.- all of them are included in reference 1.
- Armstrong, A.L.; Hunter, J.B.; Davis, T.R. The prevalence of degenerative arthritis of the base of the thumb in post-menopausal women. J Hand Surg Br 1994, 19, 340-1.
It is never clearly stated that the subjects were >65 years of age at the time of surgery. Please do so.
Answer.- en lines 164-166 of the modified manuscript is defined this subject with more detail as you sugest
Did I miss the definition of “CTS” in line 179?
Answer.- It have corrected in the mopdified manuscript to "carpal tunnel syndrome"
The authors call out the frequency of small trapeziums in the failed group, but not in the successful group (lines 266-268).
Answer.- In line 268 is refered (9 out of 50) in functional outcomes. We change functional outcomes to "(9 out of 50) in functional group"
Line 20: patient cohort
Line 22: Kaplan-Meier
Line 89-90: show acceptable long-term survival
Line 161: considered
Lines 200-202: Please correct grammar (e.g. put VAS ranges in parentheses)
Answer.- All of these grammar corrections have been made in the modified manuscript
Tank you very much for your time and for your interest